# SALD: SIGN AGNOSTIC LEARNING WITH DERIVATIVES

**Matan Atzmon & Yaron Lipman**
Weizmann Institute of Science
{matan.atzmon,yaron.lipman}@weizmann.ac.il

## ABSTRACT

Learning 3D geometry directly from raw data, such as point clouds, triangle soups, or unoriented meshes is still a challenging task that feeds many downstream computer vision and graphics applications.

In this paper, we introduce SALD: a method for learning implicit neural representations of shapes directly from raw data. We generalize sign agnostic learning (SAL) to include derivatives: given an unsigned distance function to the input raw data, we advocate a novel sign agnostic regression loss, incorporating both point-wise values and gradients of the unsigned distance function. Optimizing this loss leads to a *signed* implicit function solution, the zero level set of which is a high quality and valid manifold approximation to the input 3D data. The motivation behind SALD is that incorporating derivatives in a regression loss leads to a lower sample complexity, and consequently better fitting. In addition, we provide empirical evidence, as well as theoretical motivation in 2D that SAL enjoys a minimal surface property, favoring minimal area solutions. More importantly, we are able to show that this property still holds for SALD, i.e., with derivatives included.

We demonstrate the efficacy of SALD for shape space learning on two challenging datasets: ShapeNet (Chang et al., 2015) that contains inconsistent orientation and non-manifold meshes, and D-Faust (Bogo et al., 2017) that contains raw 3D scans (triangle soups). On both these datasets, we present state-of-the-art results.

## 1 INTRODUCTION

Recently, neural networks (NN) have been used for representing and reconstructing 3D surfaces. Current NN-based 3D learning approaches differ in two aspects: the choice of surface representation, and the supervision method. Common representations of surfaces include using NN as parametric charts of surfaces (Groueix et al., 2018b; Williams et al., 2019); volumetric implicit function representation defined over regular grids (Wu et al., 2016; Tatarchenko et al., 2017; Jiang et al., 2020); and NN used directly as volumetric implicit functions (Park et al., 2019; Mescheder et al., 2019; Atzmon et al., 2019; Chen & Zhang, 2019), referred henceforth as *implicit neural representations*. Supervision methods include regression of known or approximated volumetric implicit representations (Park et al., 2019; Mescheder et al., 2019; Chen & Zhang, 2019), regression directly with raw 3D data (Atzmon & Lipman, 2020; Gropp et al., 2020; Atzmon & Lipman, 2020), and differentiable rendering using 2D data (i.e., images) supervision (Niemeyer et al., 2020; Liu et al., 2019; Saito et al., 2019; Yariv et al., 2020).

The goal of this paper is to introduce SALD, a method for learning implicit neural representations of surfaces directly from *raw 3D data*. The benefit in learning directly from raw data, e.g., non-oriented point clouds or triangle soups (e.g., Chang et al. (2015)) and raw scans (e.g., Bogo et al. (2017)), is avoiding the need for a ground truth signed distance representation of all train surfaces for supervision. This allows working with complex models with inconsistent normals and/or missing parts. In Figure 1 we show reconstructions of zero level sets of SALD learned implicit neural representations of car models from the ShapeNet dataset (Chang et al., 2015) with variational auto-encoder; notice the high detail level and the interior, which would not have been possible with, e.g., previous data pre-processing techniques using renderings of visible parts (Park et al., 2019).

Our approach improves upon the recent Sign Agnostic Learning (SAL) method (Atzmon & Lipman, 2020) and shows that incorporating *derivatives* in a sign agnostic manner provides a significant

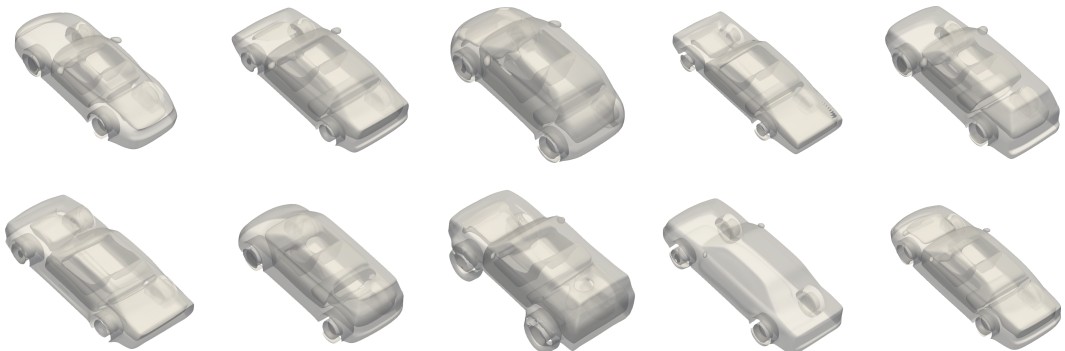

Figure 1: Learning the shape space of ShapeNet (Chang et al., 2015) cars directly from raw data using SALD. Note the interior details; top row depicts SALD reconstructions of train data, and bottom row SALD reconstructions of test data.

improvement in surface approximation and detail. SAL is based on the observation that given an unsigned distance function $h$ to some raw 3D data $\mathcal{X} \subset \mathbb{R}^3$, a sign agnostic regression to $h$ will introduce new local minima that are *signed* versions of $h$; in turn, these signed distance functions can be used as implicit representations of the underlying surface. In this paper we show how the sign agnostic regression loss can be extended to compare both function values $h$ and *derivatives* $\nabla h$, up to a sign.

The main motivation for performing NN regression with derivatives is that it reduces the *sample complexity* of the problem (Czarnecki et al., 2017), leading to better accuracy and generalization. For example, consider a one hidden layer NN of the form $f(x) = \max\{ax, bx\} + c$. Prescribing two function samples at $\{-1, 1\}$ are not sufficient for uniquely determining $f$, while adding derivative information at these points determines $f$ uniquely.

We provide empirical evidence as well as theoretical motivation suggesting that both SAL and SALD possess the favorable minimal surface property (Zhao et al., 2001), that is, in areas of missing parts and holes they will prefer zero level sets with minimal area. We justify this property by proving that, in 2D, when restricted to the zero level-set (a curve in this case), the SAL and SALD losses would encourage a straight line solution connecting neighboring data points.

We have tested SALD on the dataset of man-made models, ShapeNet (Chang et al., 2015), and human raw scan dataset, D-Faust (Bogo et al., 2017), and compared to state-of-the-art methods. In all cases we have used the raw input data $\mathcal{X}$ as is and considered the unsigned distance function to $\mathcal{X}$, i.e., $h_{\mathcal{X}}$, in the SALD loss to produce an approximate signed distance function in the form of a neural network. Comparing to state-of-the-art methods we find that SALD achieves superior results on this dataset. On the D-Faust dataset, when comparing to ground truth reconstructions we report state-of-the-art results, striking a balance between approximating details of the scans and avoiding overfitting noise and ghost geometry.

Summarizing the contributions of this paper:

- Introducing sign agnostic learning with derivatives.
- Identifying and providing a theoretical justification for the minimal surface property of sign agnostic learning in 2D.
- Training directly on raw data (end-to-end) including unoriented or not consistently oriented triangle soups and raw 3D scans.

## 2 PREVIOUS WORK

Learning 3D shapes with neural networks and 3D supervision has shown great progress recently. We review related works, where we categorize the existing methods based on their choice of 3D surface representation.

**Parametric representations.** The most fundamental surface representation is an *atlas*, that is a collection of parametric charts $f : \mathbb{R}^2 \to \mathbb{R}^3$ with certain coverage and transition properties (Do Carmo, 2016). Groueix et al. (2018b) adapted this idea using neural network to represent a surface as union of such charts; Williams et al. (2019) improved this construction by introducing better transitions between charts; Sinha et al. (2016) use geometry images (Gu et al., 2002) to represent an entire shape using a single chart; Maron et al. (2017) use global conformal parameterization for learning surface data; Ben-Hamu et al. (2018) use a collection of overlapping global conformal charts for human-shape generative model. Hanocka et al. (2020) shrink-wraps a template mesh to fits a point cloud. The benefit in parametric representations is in the ease of sampling the learned surface (i.e., forward pass) and work directly with raw data (e.g., Chamfer loss); their main struggle is in producing charts that are collectively consistent, of low distortion, and covering the shape.

**Implicit representations.** Another approach for representing surfaces is as zero level sets of a function, called an *implicit function*. There are two popular methods to model implicit volumetric functions with neural networks: i) *Convolutional neural network* predicting scalar values over a predefined fixed volumetric structure (e.g., grid or octree) in space (Tatarchenko et al., 2017; Wu et al., 2016); and ii) *Multilayer Perceptron* of the form $f : \mathbb{R}^3 \to \mathbb{R}$ defining a continuous volumetric function (Park et al., 2019; Mescheder et al., 2019; Chen & Zhang, 2019). Currently, neural networks are trained to be implicit function representations with two types of supervision: (i) regression of samples taken from a known or pre-computed implicit function representation such as occupancy function (Mescheder et al., 2019; Chen & Zhang, 2019) or a signed distance function (Park et al., 2019); and (ii) working with raw 3D supervision, by particle methods relating points on the level sets to the model parameters (Atzmon et al., 2019), using sign agnostic losses (Atzmon & Lipman, 2020), or supervision with PDEs defining signed distance functions (Gropp et al., 2020).

**Primitives.** Another type of representation is to learn shapes as composition or unions of a family of *primitives*. Gradient information have been used to improve and facilitate fitting of invariant polynomial representations (Tasdizen et al., 1999; Birdal et al., 2019). Li et al. (2019) represent a shape using a parametric collection of primitives. Genova et al. (2019; 2020) use a collection of Gaussians and learn consistent shape decompositions. Chen et al. (2020) suggest a differentiable Binary Space Partitioning tree (BSP-tree) for representing shapes. Deprelle et al. (2019) combine points and charts representations to learn basic shape structures. Deng et al. (2020) represent a shape as a union of convex sets. Williams et al. (2020) learn cites of Voronoi cells for implicit shape representation.

**Template fitting.** Lastly, several methods learn 3D shapes of a certain class (e.g., humans) by learning the deformation from a template model. Classical methods use matching techniques and geometric loss minimization for non-rigid template matching (Allen et al., 2002; 2003; Anguelov et al., 2005). Groueix et al. (2018a) use an auto-encoder architecture and Chamfer distance to match target shapes. Litany et al. (2018) use graph convolutional autoencoder to learn deformable template for shape completion.

## 3 METHOD

Given raw geometric input data $\mathcal{X} \subset \mathbb{R}^3$, e.g., a triangle soup, our goal is to find a multilayer perceptron (MLP) $f : \mathbb{R}^3 \times \mathbb{R}^m \to \mathbb{R}$ whose zero level-set,

$$\mathcal{S} = \left\{ \boldsymbol{x} \in \mathbb{R}^3 \mid f(\boldsymbol{x}; \theta) = 0 \right\} \tag{1}$$

is a manifold surface that approximates $\mathcal{X}$.

**Sign agnostic learning.** Similarly to SAL, our approach is to consider the (readily available) *unsigned* distance function to the raw input geometry,

$$h(\boldsymbol{y}) = \min_{\boldsymbol{x} \in \mathcal{X}} \|\boldsymbol{y} - \boldsymbol{x}\| \tag{2}$$

and perform sign agnostic regression to get a *signed* version $f$ of $h$. SAL uses a loss of the form

$$\text{loss}(\theta) = \mathbb{E}_{\boldsymbol{x} \sim \mathcal{D}} \, \tau \big( f(\boldsymbol{x}; \theta), h(\boldsymbol{x}) \big), \tag{3}$$

where $\mathcal{D}$ is some probability distribution, e.g., a sum of gaussians with centers uniformly sampled over the input geometry $\mathcal{X}$, and $\tau$ is an unsigned similarity. That is, $\tau(a, b)$ is measuring the difference between scalars $a, b \in \mathbb{R}$ up-to a sign. For example

$$\tau(a, b) = \big| |a| - b \big| \tag{4}$$

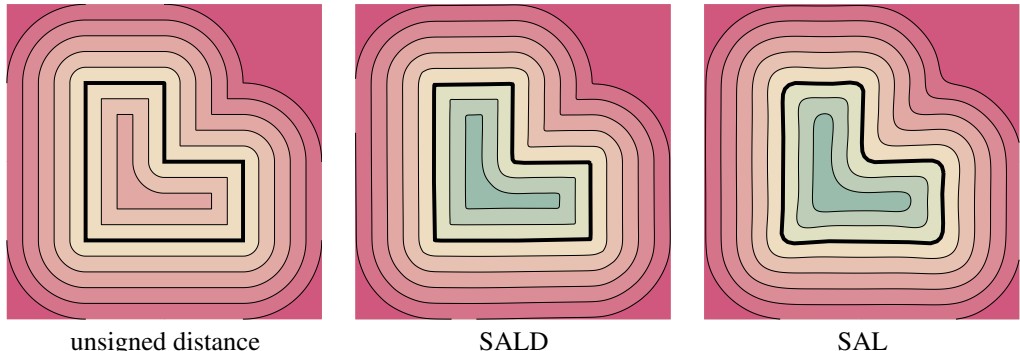

| unsigned distance | SALD | SAL |

Figure 2: Sign agnostic learning of an unsigned distance function to an L shape (left). Red colors depict positive values, and blue-green colors depict negative values. In the middle, the result of optimizing the SALD loss (equation 6); on the right, the result of SAL loss (equation 3). Note that SALD better preserves sharp features of the shape and the isolevels.

is an example that is used in Atzmon & Lipman (2020). The key property of the sign agnostic loss in equation 3 is that, with proper weights initialization $\theta_0$, it finds a new *signed* local minimum $f$ which in absolute value is similar to $h$. In turn, the zero level set $\mathcal{S}$ of $f$ is a valid manifold describing the data $\mathcal{X}$.

**Sign agnostic learning with derivatives.** Our goal is to generalize the SAL loss (equation 3) to include derivative data of $h$ and show that optimizing this loss provides implicit neural representations, $\mathcal{S}$, that enjoy better approximation properties with respect to the underlying geometry $\mathcal{X}$.

Generalizing equation 3 requires designing an unsigned similarity measure $\tau$ for vector valued functions. The key observation is that equation 4 can be written as $\tau(a, b) = \min\{|a - b|, |a + b|\}$, $a, b \in \mathbb{R}$, and can be generalized to vectors $\boldsymbol{a}, \boldsymbol{b} \in \mathbb{R}^d$ by

$$\tau(\boldsymbol{a}, \boldsymbol{b}) = \min\{\|\boldsymbol{a} - \boldsymbol{b}\|, \|\boldsymbol{a} + \boldsymbol{b}\|\}. \tag{5}$$

We define the SALD loss:

$$\text{loss}(\theta) = \mathbb{E}_{\boldsymbol{x} \sim \mathcal{D}} \, \tau\big(f(\boldsymbol{x}; \theta), h(\boldsymbol{x})\big) + \lambda \mathbb{E}_{\boldsymbol{x} \sim \mathcal{D}'} \, \tau\big(\nabla_{\boldsymbol{x}} f(\boldsymbol{x}; \theta), \nabla_{\boldsymbol{x}} h(\boldsymbol{x})\big) \tag{6}$$

where $\lambda > 0$ is a parameter, $\mathcal{D}'$ is a probability distribution, e.g., it could be identical to $\mathcal{D}$, or uniform over the input geometry $\mathcal{X}$, and $\nabla_{\boldsymbol{x}} f(\boldsymbol{x}; \theta), \nabla_{\boldsymbol{x}} h(\boldsymbol{x})$ are the gradients $f, h$ (resp.) with respect to their input $\boldsymbol{x}$.

In Figure 2 we show the unsigned distance $h$ to an L-shaped curve (left), and the level sets of the MLPs optimized with the SALD loss (middle) and the SAL loss (right); note that SALD loss reconstructed the sharp features (i.e., corners) of the shape and the level sets of $h$, while SAL loss smoothed them out; the implementation details of this experiment can be found in Appendix A.4.

**Minimal surface property.** We show that the SAL and SALD losses possess a *minimal surface property* (Zhao et al., 2001), that is, they strive to minimize surface area of missing parts. For example, Figure 4 shows the unsigned distance to a curve with a missing segment (left), and the zero level sets of MLPs optimized with SALD loss (middle), and SAL loss (right).

Note that in both cases the zero level set in the missing part area is the minimal length curve (i.e., a line) connecting the end points of that missing part. SALD also preserves sharp features of the rest of the shape. Figure A1 in the supplementary shows additional 2D experiments comparing to the Implicit Geometric Regularization (IGR) method (Gropp et al., 2020) that learns implicit representations by regularizing the gradient norm and do not posses the minimal surface property.

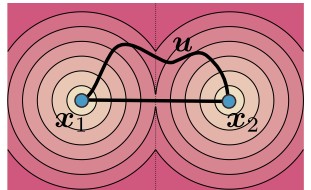

Figure 3: Minimal surface property in 2D.

We will provide a theoretical justification to this property in the 2D case. We consider a geometry defined by two points in the plane, $\mathcal{X} = \{\boldsymbol{x}_1, \boldsymbol{x}_2\} \subset \mathbb{R}^2$ and possible solutions where the zero level set curve $\mathcal{S}$ is connecting $\boldsymbol{x}_1$ and $\boldsymbol{x}_2$. We prove that among a class of

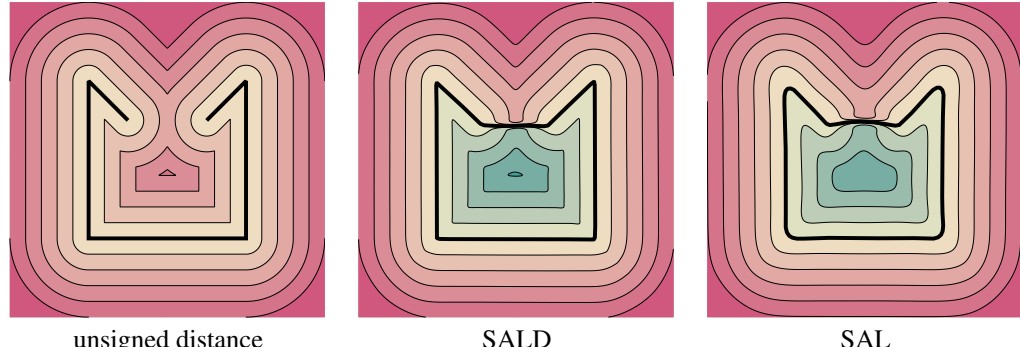

| unsigned distance | SALD | SAL |

Figure 4: Minimal surface property: using SALD (middle) and SAL (right) with the input unsigned distance function of a curve with a missing part (left) leads to a solution (black line, middle and right) with approximately minimal length in the missing part area. Note that the SALD solution also preserves sharp features of the original shape, better than SAL.

curves $\mathcal{U}$ connecting $\boldsymbol{x}_1$ and $\boldsymbol{x}_2$, the straight line minimizes the losses in equation 3 and equation 6 restricted to $\mathcal{U}$, when assuming uniform distributions $\mathcal{D}, \mathcal{D}'$. We assume (without losing generality) that $\boldsymbol{x}_1 = (0,0)^T$, $\boldsymbol{x}_2 = (\ell,0)^T$ and consider curves $\boldsymbol{u} \in \mathcal{U}$ defined by $\boldsymbol{u}(s) = (s, t(s))^T$, where $s \in [0, \ell]$, and $t : \mathbb{R} \to \mathbb{R}$ is some differentiable function such that $t(0) = 0 = t(\ell)$, see Figure 3.

For the SALD loss we prove the claim for a slightly simplified agnostic loss motivated by the following lemma proved in Appendix A.1:

**Lemma 1.** *For any pair of unit vectors $\boldsymbol{a}, \boldsymbol{b}$: $\min \{\|\boldsymbol{a} - \boldsymbol{b}\|, \|\boldsymbol{a} + \boldsymbol{b}\|\} \geq |\sin \angle(\boldsymbol{a}, \boldsymbol{b})|$.*

We consider $\tau(\boldsymbol{a}, \boldsymbol{b}) = |\sin \angle(\boldsymbol{a}, \boldsymbol{b})|$ for the derivative part of the loss in equation 6, which is also sign agnostic.

**Theorem 1.** *Let $\mathcal{X} = \{\boldsymbol{x}_1, \boldsymbol{x}_2\} \subset \mathbb{R}^2$, and the family of curves $\mathcal{U}$ connecting $\boldsymbol{x}_1$ and $\boldsymbol{x}_2$. Furthermore, let $\text{loss}_{SAL}(\boldsymbol{u})$ and $\text{loss}_{SALD}(\boldsymbol{u})$ denote the losses in equation 3 and equation 6 (resp.) when restricted to $\boldsymbol{u}$ with uniform distributions $\mathcal{D}, \mathcal{D}'$. Then in both cases the straight line, i.e., the curve $\boldsymbol{u}(s) = (s, 0)$, is the strict global minimizer of these losses.*

*Proof.* The unsigned distance function is

$$h(\boldsymbol{u}) = \begin{cases} \sqrt{s^2 + t^2} & s \in [0, \ell/2] \\ \sqrt{(s-\ell)^2 + t^2} & s \in (\ell/2, \ell] \end{cases}.$$

From symmetry it is enough to consider only the first half of the curve, i.e., $s \in [0, \ell/2)$. Then, the SAL loss, equation 3, restricted to the curve $\boldsymbol{u}$ (i.e., where $f$ vanishes) takes the form

$$\text{loss}_{SAL}(\boldsymbol{u}) = \int_0^{\ell/2} \tau(f(\boldsymbol{u}; \theta), h(\boldsymbol{u})) \|\dot{\boldsymbol{u}}\| \, ds = \int_0^{\ell/2} \sqrt{s^2 + t^2} \sqrt{1 + \dot{t}^2} \, ds,$$

where $\sqrt{1 + \dot{t}^2} \, ds$ is the length element on the curve $\boldsymbol{u}$, and $\tau(f(s, t; \theta), h(s, t)) = |h(s, t)| = \sqrt{s^2 + t^2}$, since $f(s, t; \theta) = 0$ over the curve $\boldsymbol{u}$. Plugging $t(s) \equiv 0$ in $\text{loss}_{SAL}(\boldsymbol{u})$ we see that the curve $\boldsymbol{u} = (s, 0)^T$, namely the straight line curve from $\boldsymbol{x}_1$ to $0.5(\boldsymbol{x}_1 + \boldsymbol{x}_2)$ is a strict global minimizer of $\text{loss}_{SAL}(\boldsymbol{u})$. Similar argument on $s \in [\ell/2, \ell]$ prove the claim for the SAL case.

For the SALD case, we want to calculate $\tau(\nabla_{\boldsymbol{x}} f(\boldsymbol{u}; \theta), \nabla_{\boldsymbol{x}} h(\boldsymbol{u}))$ restricted to the curve $\boldsymbol{u}$; let $\boldsymbol{a} = \nabla_{\boldsymbol{x}} f(\boldsymbol{u}; \theta)$ and $\boldsymbol{b} = \nabla_{\boldsymbol{x}} h(\boldsymbol{u})$. First, $\boldsymbol{b} = (s^2 + t^2)^{-1/2}(s, t)^T$. Second, $\boldsymbol{a}$ is normal to the curve $\boldsymbol{u}$, therefore it is proportional to $\dot{\boldsymbol{u}}^{\perp} = (-\dot{t}, 1)^T$. Next, note that

$$|\sin \angle(\boldsymbol{a}, \boldsymbol{b})| = \frac{\left| \det \begin{pmatrix} -\dot{t} & s \\ 1 & t \end{pmatrix} \right|}{\sqrt{1 + \dot{t}^2} \sqrt{s^2 + t^2}} = \frac{1}{\sqrt{1 + \dot{t}^2}} \left| \frac{d}{ds} \|(s, t)\| \right|,$$

where the last equality can be checked by differentiating $\|(s,t)\|$ w.r.t. $s$. Therefore,

$$\frac{\text{loss}_{\text{SALD}}(\boldsymbol{u}) - \text{loss}_{\text{SAL}}(\boldsymbol{u})}{\lambda} = \int_0^{\ell/2} \tau(\boldsymbol{a}, \boldsymbol{b}) \|\dot{\boldsymbol{u}}\| \, ds = \int_0^{\ell/2} \left| \frac{d}{ds} \|(s,t)\| \right| ds \geq \left\| \left( \frac{\ell}{2}, t\left(\frac{\ell}{2}\right) \right) \right\| \geq \frac{\ell}{2}.$$

This bound is achieved for the curve $\boldsymbol{u} = (s,0)$, which is also a minimizer of the SAL loss. The straight line also minimizes this version of the SALD loss since $\text{loss}_{\text{SALD}}(\boldsymbol{u}) = (\text{loss}_{\text{SALD}}(\boldsymbol{u}) - \text{loss}_{\text{SAL}}(\boldsymbol{u})) + \text{loss}_{\text{SAL}}(\boldsymbol{u})$. $\qquad\square$

## 4 EXPERIMENTS

We tested SALD on the task of shape space learning from raw 3D data. We experimented with two different datasets: i) ShapeNet dataset (Chang et al., 2015), containing synthetic 3D Meshes; and ii) D-Faust dataset (Bogo et al., 2017) containing raw 3D scans. Furthermore, we empirically test our sample complexity hypothesis (i.e., that incorporating derivatives improve sample complexity) by inspecting surface reconstruction accuracy for SAL and SALD when trained with fixed size sample sets.

**Shape space learning architecture.** Our method can be easily incorporated into existing shape space learning architectures: i) Auto-Decoder (AD) suggested in Park et al. (2019); and the ii) Modified Variational Auto-Encoder (VAE) used in Atzmon & Lipman (2020). For VAE, the encoder is taken to be PointNet (Qi et al., 2017). For both options, the decoder is the implicit representation in equation 1, where $f(\boldsymbol{x}; \theta)$ is taken to be an 8-layer MLP with 512 hidden units in each layer and Softplus activation. In addition, to enable sign agnostic learning we initialize the decoder weights, $\theta$, using the geometric initialization from Atzmon & Lipman (2020). See Appendix A.2.4 for more details regarding the architecture. The point samples $\boldsymbol{x}, \boldsymbol{x}'$ for the empirical computation of the expectations in equation 6 are drawn according to distributions $\mathcal{D}, \mathcal{D}'$ explained in Appendix A.2.1.

**Baselines.** The baseline methods selected for comparison cover both existing supervision methodologies: DeepSDF (Park et al., 2019) is chosen as a representative out of the methods that require pre-computed implicit representation for training. For methods that train directly on raw 3D data, we compare versus SAL (Atzmon & Lipman, 2020) and IGR (Gropp et al., 2020). See Appendix A.6 for a detailed description of the quantitative metrics used for evaluation.

| Category | Sofas | | Chairs | | Tables | | Planes | | Lamps | |
|---|---|---|---|---|---|---|---|---|---|---|
| | Mean | Median | Mean | Median | Mean | Median | Mean | Median | Mean | Median |
| DeepSDF | **0.329** | **0.230** | **0.341** | 0.133 | 0.839 | **0.149** | **0.177** | 0.076 | **0.909** | **0.344** |
| SAL | 0.704 | 0.523 | 0.494 | 0.259 | **0.543** | **0.231** | 0.429 | 0.146 | 4.913 | 1.515 |
| SALD(VAE) | 0.391 | 0.244 | 0.415 | 0.255 | 0.679 | 0.279 | 0.197 | **0.062** | 1.808 | 1.172 |
| SALD(AD) | **0.207** | **0.147** | **0.281** | **0.157** | **0.408** | 0.25 | **0.098** | **0.032** | **0.506** | **0.327** |

Table 1: ShapeNet quantitative results. We log the mean and median of the Chamfer distances ($d_C$) between the reconstructed 3D surfaces and the ground truth meshes. Numbers are reported $*10^3$.

### 4.1 SHAPENET

In this experiment we tested the ability of SALD to learn a shape space by training on a challenging 3D data such as non-manifold/non-orientable meshes. We tested SALD with both AD and VAE architectures. In both settings, we set $\lambda = 0.1$ for the SALD loss. We follow the evaluation protocol as in DeepSDF (Park et al., 2019): using

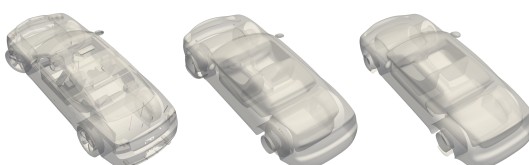

Figure 5: AD versus VAE.

the same train/test splits, we train and evaluate our method on 5 different categories. Note that comparison versus IGR is omitted as IGR requires consistently oriented normals for shape space learning, which is not available for ShapeNet, where many models have non-consistent triangles' orientation.

**Results.** Table 1 and Figure 6 show quantitative and qualitative results (resp.) for the held-out test set, comparing SAL, DeepSDF and SALD. As can be read from the table and inspected in the

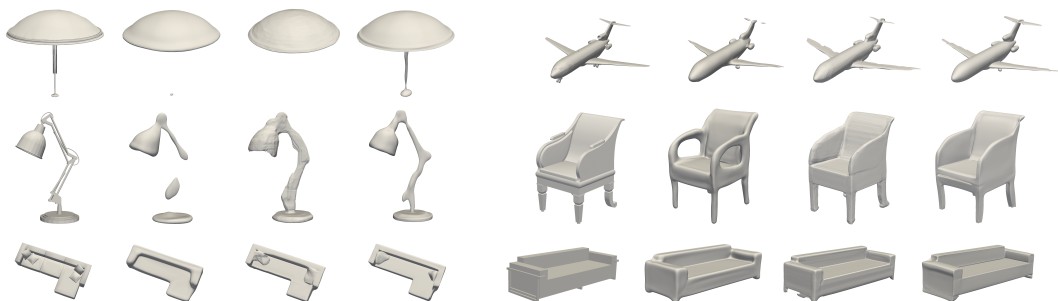

Figure 6: ShapeNet qualitative test results. Each quadruple shows (columns from left to right): ground truth model, SAL-reconstruction, DeepSDF reconstruction, SALD reconstruction.

figure, our method, when used with the same auto-decoder as in DeepSDF, compares favorably to DeepSDF's reconstruction performance on this data.

Qualitatively the surfaces produced by SALD are smoother, mostly with more accurate sharp features, than SAL and DeepSDF generated surfaces. Figure 1 shows typical train and test results from the Cars class with VAE. Figure 5 shows a comparison between SALD shape space learning with VAE and AD in reconstruction of a test car model (left). Note that the AD (middle) seems to produce more details of the test model than the VAE (right), e.g., steering wheel and headlights. Figure 7 show SALD (AD) generated shapes via latent space interpolation between two test models.

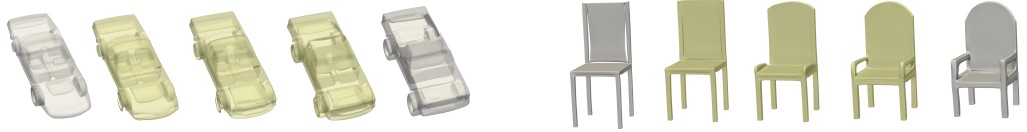

Figure 7: ShapeNet latent interpolation. In each group, the leftmost and rightmost columns are test examples reconstructions; latent space generated shapes are coloured in yellow.

## 4.2 D-FAUST

The D-Faust dataset (Bogo et al., 2017) contains raw scans (triangle soups) of 10 humans in multiple poses. There are approximately 41k scans in the dataset. Due to the low variety between adjacent scans, we sample each pose scans at a ratio of $1:5$. The leftmost column in Figure 8 shows examples of raw scans used for training. For evaluation we use the *registrations* provided with the data set. Note that the registrations where not used for training. We tested SALD using the VAE architecture, with $\lambda = 1.0$ set for the SALD loss. We followed the evaluation protocol as in Atzmon & Lipman (2020), using the same train/test split. Note that Atzmon & Lipman (2020) already conducted a comprehensive comparison of SAL versus DeepSDF and AtlasNet (Groueix et al., 2018b), establishing SAL as a state-of-the-art method for this dataset. Thus, we focus on comparison versus SAL and IGR.

**Results.** Table 2 and Figure 8 show quantitative and qualitative results (resp.); although SALD does not produces the best test quantitative results, it is roughly comparable in every measure to the best among the two baselines. That is, it produces details comparable to IGR while maintaining the minimal surface property as SAL and not adding undesired surface sheets as IGR; see the figure for visual illustrations of these properties: the high level of details of SALD and IGR compared to SAL, and the extraneous parts added by IGR, avoided by SALD. These phenomena can also be seen quantitatively, e.g., the reconstruction-to-registration loss of IGR. Figure 9 show SALD generated shapes via latent space interpolation between two test scans. Notice the ability of SALD to generate novel mixed faces and body parts.

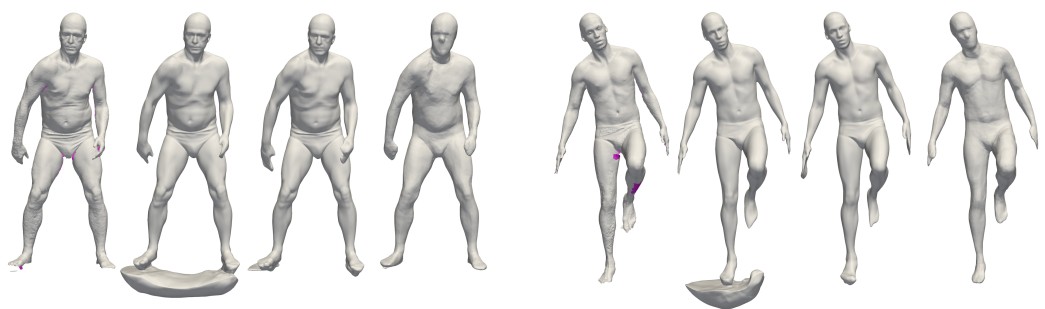

Figure 8: D-Faust qualitative results on test examples. Each quadruple shows (columns from left to right): raw scans (magenta depict back-faces), IGR, SALD, and SAL.

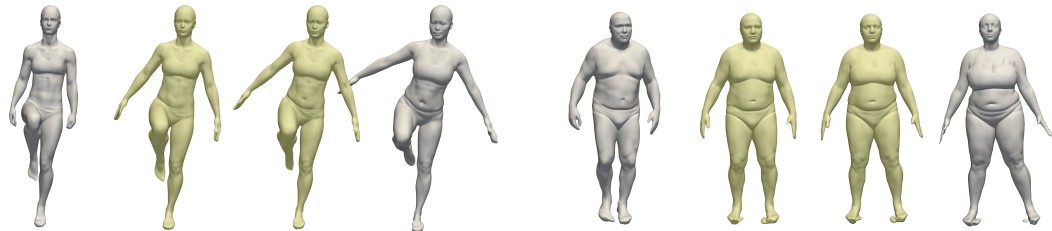

Figure 9: D-Faust latent interpolation. In each group, the leftmost and rightmost columns are test scans reconstructions; latent space generated shapes are coloured in yellow.

### 4.3 SAMPLE COMPLEXITY

In this experiment we test the sample complexity hypothesis: namely, whether regressing with derivatives improves shape reconstruction accuracy, under a *fixed budget* of point samples. This experiment considers 3 different shapes chosen randomly from the chair, sofa and table test sets of the ShapeNet dataset. For each

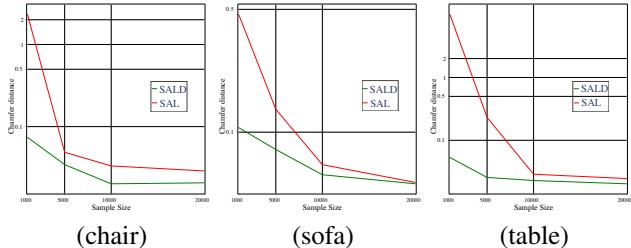

(chair)      (sofa)      (table)

shape, we prepared a *fixed* sample set of points $\{x_i\}_{i=1}^m$, where $m \in \{1K, 5K, 10K, 20K\}$, together with the unsigned distance value and derivative $\{h(x_i), \nabla_x h(x_i)\}_{i=1}^m$. The point samples are drawn according to distribution $\mathcal{D}$ as explained in Appendix A.2.1. We separately trained the SAL and SALD losses on the same sample data, using the same hyper-parameters, in two different scenarios: i) Individual shape reconstruction: optimizing the weights $\theta$ of a randomly initialized 8-layer MLP $f(x; \theta)$; and ii) latent shape reconstruction: given a *trained* auto-decoder network $f(x, z; \theta)$ (as in 4.1), we optimize solely the latent code $z$, keeping the weights $\theta$ fixed. Lastly, we computed the Chamfer distance between the learned shape $\mathcal{S}$ and the input geometry $\mathcal{X}$. For the individual shape reconstruction, the inset figure shows the Chamfer distance, $d_C(\mathcal{S}, \mathcal{X})$, as a function of the sample size $m$. Figure 10 shows for each sample size, the learned sofa and table. Note that SALD demonstrates better approximation to the input geometry in comparison to SAL,

| | $d_{\overrightarrow{C}}$ (reg., recon.) | | $d_{\overrightarrow{N}}$ (reg., recon.) | | $d_{\overrightarrow{C}}$ (recon., reg.) | | $d_{\overrightarrow{N}}$ (recon., reg.) | | $d_{\overrightarrow{C}}$ (scan, recon.) | | $d_{\overrightarrow{N}}$ (scan, recon.) | |
|---|---|---|---|---|---|---|---|---|---|---|---|---|
| | Mean | Median | Mean | Median | Mean | Median | Mean | Median | Mean | Median | Mean | Median |
| SAL | **0.418** | **0.328** | 13.21 | 12.459 | **0.344** | **0.256** | **11.354** | **10.522** | 0.429 | **0.246** | 10.096 | 9.096 |
| IGR | **0.276** | **0.187** | **10.328** | **9.822** | 3.806 | 3.627 | 17.124 | 17.902 | **0.241** | **0.11** | **5.829** | **5.295** |
| SALD | 0.428 | 0.346 | **11.67** | **11.07** | **0.489** | **0.362** | 11.035 | 10.371 | **0.397** | 0.279 | **7.884** | **7.227** |

Table 2: D-Faust quantitative results. We log mean and median of the one-sided Chamfer and normal distances between registration meshes (reg), reconstructions (recon) and raw input scans (scan). The $d_C$ numbers are reported $*10^2$.

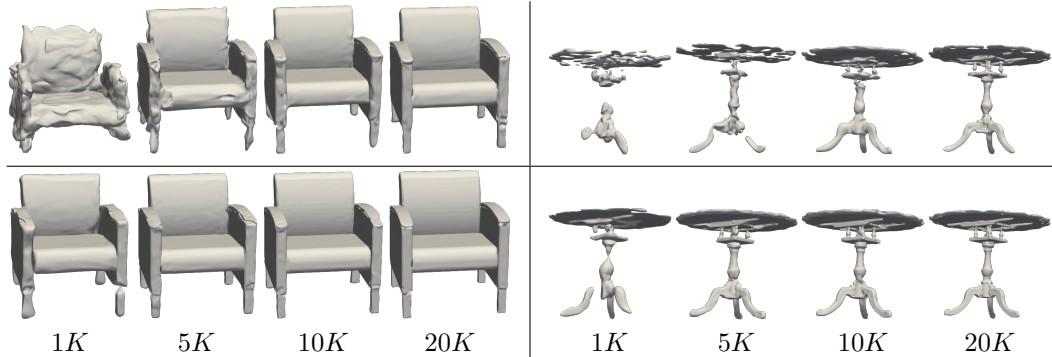

Figure 10: Sample complexity experiment: SALD (bottom row) shows better shape approximation than SAL (top row), especially for small sample sets; numbers indicate sample sizes.

in particular as the sample size gets smaller, and thus supporting the sample complexity hypothesis. When optimizing the latent code of a fully trained auto-decoder, the sample size has a little to no effect on the approximation quality of a test shape reconstruction. This can be explained by the fact that the auto-decoder is trained on the maximal sample size, and therefore provides a strong prior for the latent reconstruction. See the supplementary A.5, for the results.

### 4.4 LIMITATIONS

Figure 11 shows typical failure cases of our method from the ShapeNet experiment described above. We mainly suffer from two types of failures: First, since inside and outside information is not known (and often not even well defined in ShapeNet models) SALD can add surface sheets closing what

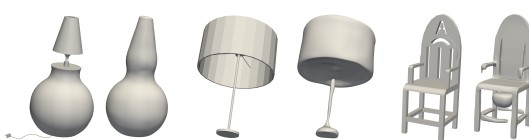

Figure 11: Failure cases.

should be open areas (e.g., the bottom side of the lamp, or holes in the chair). Second, thin structures can be missed (e.g., the electric cord of the lamp on the left). A useful strategy to sample thin structures is to make sure the sample frequency is inversely proportional to the distance to the medial axis Amenta et al. (1998), where an approximation can be made using curvature estimation. Furthermore, it is important to note that implicit representations of the type presented in equation 1 cannot model surfaces with boundaries and therefore cannot represent flat dimensionless surfaces with boundaries. A potential solution could be incorporating additional implicits to handle boundaries.

## 5 CONCLUSIONS

We introduced SALD, a method for learning implicit neural representations from raw data. The method is based on a generalization of the sign agnostic learning idea to include derivative data. We demonstrated that the addition of a sign agnostic derivative term to the loss improves the approximation power of the resulting signed implicit neural network. In particular, showing improvement in the level of details and sharp features of the reconstructions. Furthermore, we identify the favorable minimal surface property of the SAL and SALD losses and provide a theoretical justification in 2D. Generalizing this theoretical analysis to 3D is marked as interesting future work.

We see two more possible venues for future work: First, it is clear that there is room for further improvement in approximation properties of implicit neural representations. Although the results in D-Faust are already close to the input quality, in ShapeNet we still see a gap between input models and their implicit neural representations; this challenge already exists in overfitting a large collection of diverse shapes in the training stage. Improvement can come from adding expressive power to the neural networks, or further improving the training losses; adding derivatives as done in this paper is one step in that direction but does not solves the problem completely. Combining sign agnostic learning with the recent positional encoding method (Tancik et al., 2020) could also be an interesting future research venue. Another interesting project is to combine the sign-agnostic losses with gradient regularization such as the one employed in IGR (Gropp et al., 2020). Second, it is interesting to think of applications or settings in which SALD can improve the current state-of-the-art. Generative 3D modeling, learning geometry with 2D supervision, or other types of partially observed scans such as depth images are all potentially fruitful options.

ACKNOWLEDGMENTS

The research was supported by the European Research Council (ERC Consolidator Grant, "Lift-Match" 771136), the Israel Science Foundation (Grant No. 1830/17) and by a research grant from the Carolito Stiftung (WAIC).

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

# A   APPENDIX

## A.1   PROOF OF LEMMA 1

**Lemma 1.** *For any pair of unit vectors $\boldsymbol{a}, \boldsymbol{b}$:* $\min\{\|\boldsymbol{a} - \boldsymbol{b}\|, \|\boldsymbol{a} + \boldsymbol{b}\|\} \geq |\sin \angle(\boldsymbol{a}, \boldsymbol{b})|$.

*Proof.* Let $\boldsymbol{a}, \boldsymbol{b} \in \mathbb{R}^d$ be arbitrary unit norm vectors. Then,

$$
\begin{aligned}
\min\{\|\boldsymbol{a} - \boldsymbol{b}\|, \|\boldsymbol{a} + \boldsymbol{b}\|\} &= \left[\min\{2 + 2\langle \boldsymbol{a}, \boldsymbol{b}\rangle, 2 - 2\langle \boldsymbol{a}, \boldsymbol{b}\rangle\}\right]^{1/2} \\
&= \sqrt{2}\left[1 - |\langle \boldsymbol{a}, \boldsymbol{b}\rangle|\right]^{1/2} \\
&= 2\left[\frac{1 - |\cos \angle(\boldsymbol{a}, \boldsymbol{b})|}{2}\right]^{1/2} \\
&\geq |\sin \angle(\boldsymbol{a}, \boldsymbol{b})|.
\end{aligned}
$$

Where the last inequality can be proved by considering two cases: $\alpha \in [0, \pi/2]$ and $\alpha \in [\pi/2, \pi]$, where we denote $\alpha = \angle(\boldsymbol{a}, \boldsymbol{b})$. In the first case $\alpha \in [0, \pi/2]$, $\cos \alpha \geq 0$ and in this case $\sqrt{\frac{1 - \cos \alpha}{2}} = \left|\sin \frac{\alpha}{2}\right|$. The inequality is proved by considering

$$
2\left|\sin \frac{\alpha}{2}\right| - |\sin \alpha| = 2\sin \frac{\alpha}{2} - \sin \alpha = 2\sin \frac{\alpha}{2}(1 - \cos \frac{\alpha}{2}) \geq 0
$$

for $\alpha \in [0, \pi/2]$. For the case $\alpha \in [\pi/2, \pi]$ we have $\sqrt{\frac{1 + \cos \alpha}{2}} = \left|\cos \frac{\alpha}{2}\right|$. This case is proved by considering

$$
2\left|\cos \frac{\alpha}{2}\right| - |\sin \alpha| = 2\cos \frac{\alpha}{2} - \sin \alpha = 2\cos \frac{\alpha}{2}(1 - \sin \frac{\alpha}{2}) \geq 0
$$

for $\alpha \in [\pi/2, \pi]$  □

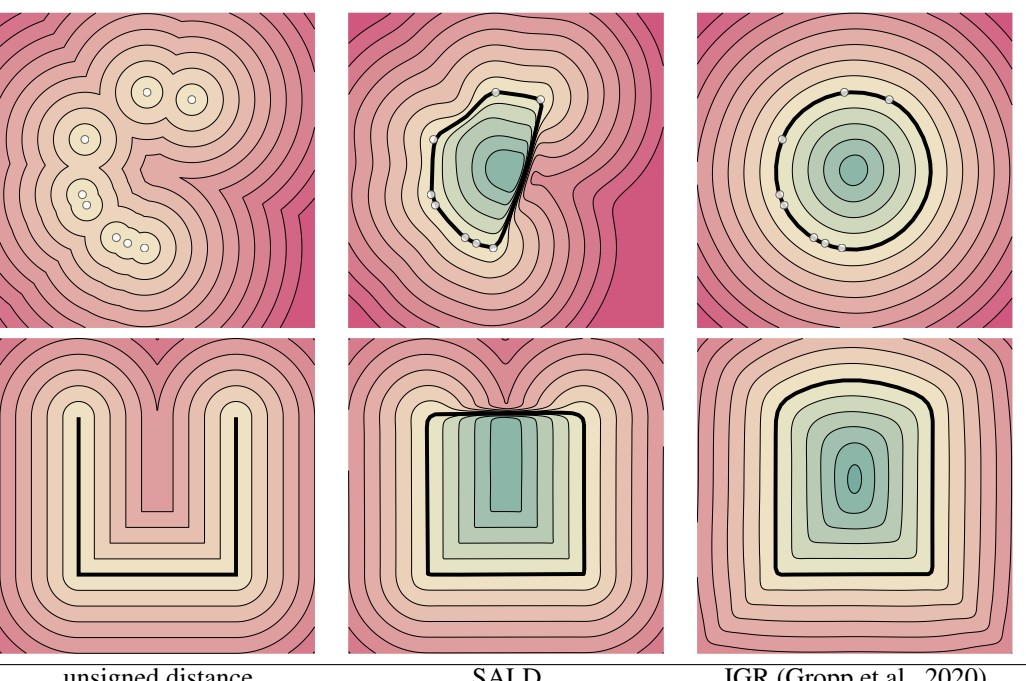

| unsigned distance | SALD | IGR (Gropp et al., 2020) |

Figure A1: 2D reconstruction additional results.

## A.2 IMPLEMENTATION DETAILS

### A.2.1 DATA PREPARATION

Given some raw 3D data $\mathcal{X}$, SALD loss (See equation 6) is computed on points and corresponding unsigned distance derivatives, $\{h(\boldsymbol{x})\}_{\boldsymbol{x}\in\mathcal{D}}$ and $\{\nabla_{\boldsymbol{x}}h(\boldsymbol{x}')\}_{\boldsymbol{x}'\in\mathcal{D}'}$ (resp.) sampled from some distributions $\mathcal{D}$ and $\mathcal{D}'$. In this paper, we set $\mathcal{D} = \mathcal{D}_1 \cup \mathcal{D}_2$, where $\mathcal{D}_1$ is chosen by uniform sampling points $\{\boldsymbol{y}\}$ from $\mathcal{X}$ and placing two isotropic Gaussians, $\mathcal{N}(\boldsymbol{y}, \sigma_1^2 I)$ and $\mathcal{N}(\boldsymbol{y}, \sigma_2^2 I)$ for each $\boldsymbol{y}$. The distribution parameter $\sigma_1$ depends on each point $\boldsymbol{y}$, set to be as the distance of the $50^{\text{th}}$ closest point to $\boldsymbol{y}$, whereas $\sigma_2$ is set to 0.3 fixed. $\mathcal{D}_2$ is chosen by projecting $\mathcal{D}_1$ to $\mathcal{S}$. The distribution $\mathcal{D}'$ is set to uniform on $\mathcal{X}$; note that on $\mathcal{X}$, $\nabla_{\boldsymbol{x}}h(\boldsymbol{x}')$ is a sub-differential which is the convex hull of the two possible normal vectors $(\pm\boldsymbol{n})$ at $\boldsymbol{x}'$; as the sign-agnostic loss does not differ between the two normal choices, we arbitrarily use one of them in the loss. Computing the unsigned distance to $\mathcal{X}$ is done using the CGAL library (The CGAL Project, 2020). To speed up training, we precomputed for each shape in the dataset, 500K samples of the form $\{h(\boldsymbol{x})\}_{\boldsymbol{x}\in\mathcal{D}}$ and $\{\nabla_{\boldsymbol{x}}h(\boldsymbol{x}')\}_{\boldsymbol{x}'\in\mathcal{D}'}$.

### A.2.2 GRADIENT COMPUTATION

The SALD loss requires incorporating the term $\nabla_{\boldsymbol{x}}f(\boldsymbol{x};\theta)$ in a differentiable manner. Our computation of $\nabla_{\boldsymbol{x}}f(\boldsymbol{x};\theta)$ is based on AUTOMATIC DIFFERENTIATION (Baydin et al., 2017) forward mode. Similarly to Gropp et al. (2020), $\nabla_{\boldsymbol{x}}f(\boldsymbol{x};\theta)$ is constructed as a network consists of layers of the form

$$\nabla_{\boldsymbol{x}}\boldsymbol{y}^{\ell+1} = \text{diag}\left(\sigma'\left(\boldsymbol{W}_{\ell+1}\boldsymbol{y}^{\ell} + \boldsymbol{b}_{\ell+1}\right)\right)\boldsymbol{W}_{\ell+1}\nabla_{\boldsymbol{x}}\boldsymbol{y}^{\ell}$$

where $\boldsymbol{y}^{\ell}$ denotes the output of the $\ell$ layer in $f(\boldsymbol{x};\theta)$ and $\boldsymbol{\theta} = (\boldsymbol{W}_{\ell}, \boldsymbol{b}_{\ell})$ are the learnable parameters.

### A.2.3 TIMINGS AND NETWORK SIZE

In figure A2, we report the timings and memory footprint of a 8-layer MLP with 512 hidden units. As the gradients calculation, $\nabla_{\boldsymbol{x}}f(\boldsymbol{x};\theta)$, is based on automatic differentiation forward mode, in theory it should yield doubling of the forward time. However, in practice we see that the gap increases as we increase the number of points for evaluation. For the DFaust experiment (which is the largest dataset in the paper), training was done a batch of 64 shapes and a sample size of $92^2$. It took around 1.5 days to complete 3000 epochs with 4 Nvidia V100 32GB gpus. Note that for VAE, the computational cost in test time is equivalent between SAL and SALD.

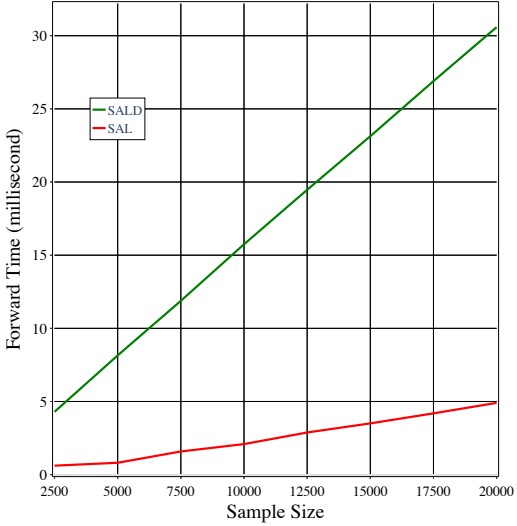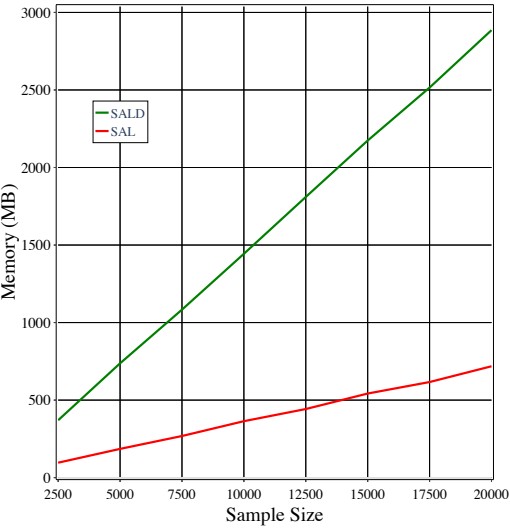

Figure A2: Timings (left) and network memory footprint (right), reported on various sample size.

### A.2.4 ARCHITECTURE DETAILS

#### VAE ARCHITECTURE

Our VAE architecture is based on the one used in Atzmon & Lipman (2020). The encoder $g\left(\boldsymbol{X};\boldsymbol{\theta}_1\right)$, where $\boldsymbol{X} \in \mathbb{R}^{N \times 3}$ is the input point cloud, is composed of DeepSets (Zaheer et al., 2017) and PointNet (Qi et al., 2017) layers. Each layer consists of

$$\text{PFC}(d_{\text{in}}, d_{\text{out}}) : \boldsymbol{X} \mapsto \nu\left(\boldsymbol{X}W + \mathbf{1}b^T\right)$$
$$\text{PL}(d_{\text{in}}, 2d_{\text{in}}) : \boldsymbol{Y} \mapsto [\boldsymbol{Y}, \max\left(\boldsymbol{Y}\right)\mathbf{1}]$$

where $[\cdot, \cdot]$ is the concat operation, $W \in \mathbb{R}^{d_{\text{in}} \times d_{\text{out}}}$ and $b \in \mathbb{R}^{d_{\text{out}}}$ are the layer weights and bias and $\nu\left(\cdot\right)$ is the pointwise non-linear ReLU activation function. Our encoder architecture is:

$$\text{PFC}(3, 128) \rightarrow \text{PFC}(128, 128) \rightarrow \text{PL}(128, 256) \rightarrow$$
$$\text{PFC}(256, 128) \rightarrow \text{PL}(128, 256) \rightarrow \text{PFC}(256, 128) \rightarrow$$
$$\text{PL}(128, 256) \rightarrow \text{PFC}(256, 128) \rightarrow \text{PL}(128, 256) \rightarrow$$
$$\text{PFC}(256, 256) \rightarrow \text{MaxPool} \overset{\times 2}{\rightarrow} \text{FC}(256, 256),$$

where $\text{FC}(d_{\text{in}}, d_{\text{out}}) : \boldsymbol{x} \mapsto \nu\left(W\boldsymbol{x} + \boldsymbol{b}\right)$ denotes a fully connected layer. The final two fully connected layers outputs vectors $\boldsymbol{\mu} \in \mathbb{R}^{256}$ and $\boldsymbol{\eta} \in \mathbb{R}^{256}$ used for parametrization of a multiviariate Gaussian $\mathcal{N}(\boldsymbol{\mu}, \text{diag}\exp\boldsymbol{\eta})$ used for sampling a latent vector $\boldsymbol{z} \in \mathbb{R}^{256}$. Our encoder architecture is similar to the one used in Mescheder et al. (2019).

Our decoder $f\left([\boldsymbol{x}, \boldsymbol{z}]; \boldsymbol{\theta}_2\right)$ is a composition of 8 layers where the first layer is $\text{FC}(256 + 3, 512)$, middle layers are $\text{FC}(512, 512)$ and the final layer is $\text{Linear}(512, 1)$. Notice that the input for the decoder is $[\boldsymbol{x}, \boldsymbol{z}]$ where $\boldsymbol{x} \in \mathbb{R}^3$ and $\boldsymbol{z}$ is the latent vector. In addition, we add a skip connection between the input to the middle fourth layer. We chose the Softplus with $\beta = 100$ for the non linear activation in the FC layers. For regulrization of the latent $\boldsymbol{z}$, we add the following term to training loss

$$0.001 * \left(\|\boldsymbol{\mu}\|_1 + \|\boldsymbol{\eta} + \mathbf{1}\|_1\right),$$

similarly to Atzmon & Lipman (2020).

#### AUTO-DECODER ARCHITECTURE

We use an auto-decoder architecture, similar to the one suggested in Park et al. (2019). We defined the latent vector $\boldsymbol{z} \in \mathbb{R}^{256}$. The decoder architecture is the same as the one described above for the VAE. For regulrization of the latent $\boldsymbol{z}$, we add the following term to the loss

$$0.001 * \|\boldsymbol{z}\|_2^2,$$

similarly to Park et al. (2019).

### A.3 TRAINING DETAILS

We trained our networks using the ADAM (Kingma & Ba, 2014) optimizer, setting the batch size to 64. On each training step the SALD loss is evaluated on a random draw of $92^2$ points out of the precomputed 500K samples. For the VAE, we set a fixed learning rate of 0.0005, whereas for the AD we scheduled the learning rate to start from 0.0005 and decrease by a factor of 0.5 every 500 epochs. All models were trained for 3000 epochs. Training was done on 4 Nvidia V-100 GPUs, using PYTORCH deep learning framework (Paszke et al., 2017).

### A.4 FIGURES 2 AND 4

For the two dimensional experiments in figures 2 and 4 we have used the same decoder as in the VAE architecture with the only difference that the first layer is $\text{FC}(2, 512)$ (no concatenation of a latent vector to the 2D input). We optimized using the ADAM (Kingma & Ba, 2014) optimizer, for 5000 epochs. The parameter $\lambda$ in the SALD loss was set to 0.1.

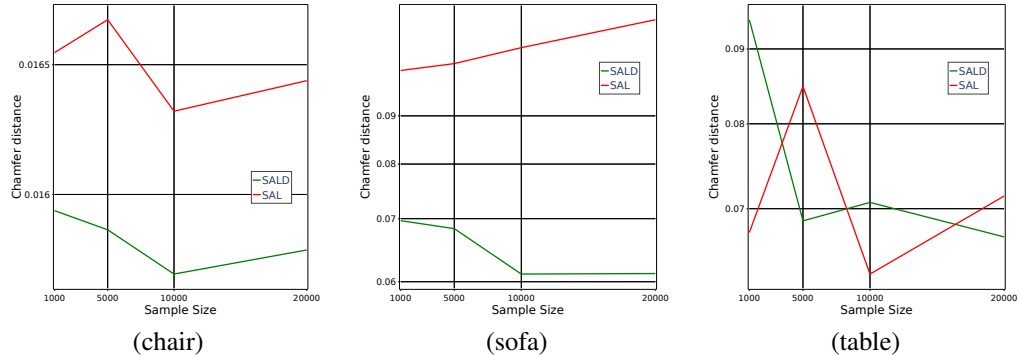

| (chair) | (sofa) | (table) |

Figure A3: Latent reconstruction sample complexity experiment: Chamfer distance to the input, as a function of the sample size. Note the Chamfer distance of the latent reconstruction is oblivious to sample size.

## A.5   SAMPLE COMPLEXITY

Figures A3 and A4 show quantitative and qualitative results for auto-decoder latent test shape reconstruction on samples of sizes $1K, 5K, 10K, 20K$. Note that the reconstruction is oblivious to the sample size. This is possibly due to the fact that the auto-decoder was trained with the maximal sample size.

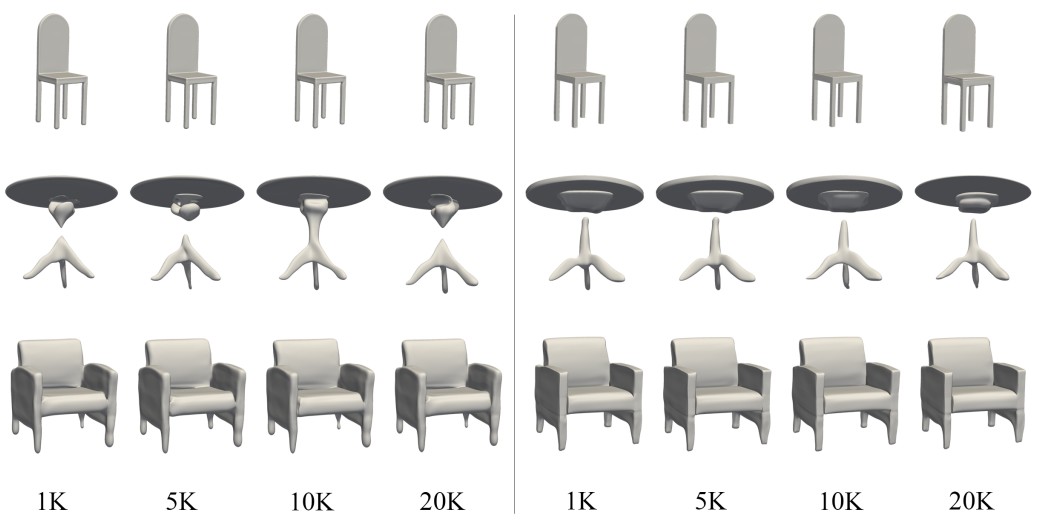

| 1K | 5K | 10K | 20K | 1K | 5K | 10K | 20K |

Figure A4: Latent reconstruction sample complexity experiment: SAL is left, SALD is right. Note the latent reconstruction is oblivious to sample size.

## A.6   EVALUATION

**Evaluation metrics.**   We use the following Chamfer distance metrics to measure similarity between shapes:

$$d_C(\mathcal{X}_1, \mathcal{X}_2) = \frac{1}{2}\left(d_C^{\rightarrow}(\mathcal{X}_1, \mathcal{X}_2) + d_C^{\rightarrow}(\mathcal{X}_2, \mathcal{X}_1)\right) \tag{7}$$

where

$$d_C^{\rightarrow}(\mathcal{X}_1, \mathcal{X}_2) = \frac{1}{|\mathcal{X}_1|}\sum_{\boldsymbol{x}_1 \in \mathcal{X}_1}\min_{\boldsymbol{x}_2 \in \mathcal{X}_2}\|\boldsymbol{x}_1 - \boldsymbol{x}_2\| \tag{8}$$

and the sets $\mathcal{X}_i$ are either point clouds or triangle soups. In addition, to measure similarity of the normals of triangle soups $\mathcal{T}_1, \mathcal{T}_2$, we define:

$$d_N (\mathcal{T}_1, \mathcal{T}_2) = \frac{1}{2} \left( \vec{d_N} (\mathcal{T}_1, \mathcal{T}_2) + \vec{d_N} (\mathcal{T}_2, \mathcal{T}_1) \right), \qquad (9)$$

where

$$\vec{d_N} (\mathcal{T}_1, \mathcal{T}_2) = \frac{1}{|\mathcal{T}_1|} \sum_{\boldsymbol{x}_1 \in \mathcal{T}_1} \angle(\boldsymbol{n}(\boldsymbol{x}_1), \boldsymbol{n}(\hat{\boldsymbol{x}}_1)), \qquad (10)$$

where $\angle(\boldsymbol{a}, \boldsymbol{b})$ is the positive angle between vectors $\boldsymbol{a}, \boldsymbol{b} \in \mathbb{R}^3$, $\boldsymbol{n}(\boldsymbol{x}_1)$ denotes the face normal of a point $\boldsymbol{x}_1$ in triangle soup $\mathcal{T}_1$, and $\hat{\boldsymbol{x}}_1$ is the projection of $\boldsymbol{x}_1$ on $\mathcal{T}_2$.

Tables 1 and 2 in the main paper report quantitative evaluation of our method, compared to other baselines. The meshing of the learned implicit representation was done using the MARCHING CUBES algorithm (Lorensen & Cline, 1987) on a uniform cubical grid of size $[512]^3$. Computing the evaluation metrics $d_C$ and $d_N$ is done on a uniform sample of 30K points from the meshed surface.

