# OpenReview forum: "SALD: Sign Agnostic Learning with Derivatives"
_ICLR.cc/2021/Conference — ICLR 2021 Poster_

### Official Review · AnonReviewer4 · 2020-10-27
**Good paper but needs some revisions for acceptance**

**Rating:** 7
**Confidence:** 4

**Review:**

## Summary of paper and contributions
SALD extends prior work on Sign Agnostic neural implicit shape representations to include a loss term on the derivative of the implicit function. The authors justify the benefits of derivatives in 2 ways: (a) By citing prior work [1] which shows empirically that derivatives decrease sample complexity of deep ReLU networks, and (b) By showing qualitative improvements over SAL without derivatives.

The authors show qualitative evidence that global minimizers of sign agnostic losses (with and without derivatives) satisfy the *minimal surface property*, a desirable property of solutions in commonly discussed the surface reconstruction literature. They demonstrate this property via 2D experiments and via a motivating theoretical example.
Finally, the authors show their loss function can be integrated into existing generative shape modelling pipelines, comparing results on ShapeNet and D-FAUST against DeepSDF which requires pre-computed SDF data, and SAL which can operate on raw inputs.

## On the benefit of using derivatives
The authors cite [1] to motivate the benefit of including derivative terms in the loss. In the case of deep ReLU networks such as the one used by the authors, this prior work shows an empirical reduction in sample complexity when regressing low dimensional functions (Section 4.1) motivated by a theoretical intuition (Section 3).  While the neural implicit functions learned by SALD are indeed low dimensional, the shape-space learning problem is not: It learns a map from a point set (consisting of many points) or a high dimensional (256 in the SALD case) latent code to an implicit function. Given this, I don't believe the authors can simply claim a reduction in sample complexity by citing [1] without demonstrating further experimental evidence, especially given the fact that the experiments in the paper do not show SALD drastically improving over SAL.

In particular I would be more convinced by an experiment showing the degradation of SAL vs SALD as the number of available samples for a shape is decreased when (a) regressing a single shape directly from data (such as in IGR [2] Section 6), and (b) regressing a shape using an auto-decoder.

## Minimal surface property
Showing that global minima to SAL may satisfy the minimal surface property is indeed quite interesting. I do feel however that the claim in the paper regarding this is a bit oversold. In particular "We prove that SAL enjoys a minimal length property in 2D" (Abstract) and "Identifying and providing a theoretical justification for the minimal surface property of [sal]." (end of Section 1). The minimal surface property is well known in the surface reconstruction literature (e.g. [3] cited by the authors in Section 3) and the theorem shown by the authors appears to be for a specific case in 2D unless I am missing something. While these results are not trivial, I feel the contribution should be rephrased to something along the lines of
"We give empirical evidence and theoretical motivation that minimizers of SAL-type losses produce solutions satisfying the minimal surface property [citation]"

## Experimental Evidence
I feel like the choices of datasets and baselines are sufficient to show the effectiveness of SALD. There are two experiments however which I feel are missing from the paper:
 1. The sample complexity experiment described above.
 2. Some kind of performance evaluation. I imagine that computing losses on gradients of networks is quite expensive. How much is the increase in runtime compared to the gains in accuracy?

## Summary of review
Generalizing SAL to include derivative quantities is a natural next step for this line of work. The authors show that SALD improves performance over the state of the art on Shapenet and performs comparably on D-FAUST. While these results are great, I feel the paper is missing a few key experiments described above, and that the claims around the minimal surface property are a bit overblown. I am rating this paper as marginally below the acceptance threshold but am more than willing to increase my score if the authors make the requested revisions or give a strong justification as to why they are unnecessary in their rebuttal.

## References
[1] Czarnecki et. al. - Sobolev Training for Neural Networks
[2] Gropp e.t. al. - Implicit Geometric Regularization for Learning Shapes
[3] Zhao et. al. - Fast surface reconstruction using the level set method

---

> ### Author Response · Authors · 2020-11-20
> **Response to reviewer4**
>
> **Q: In particular I would be more convinced by an experiment showing the degradation of SAL vs SALD as the number of available samples for a shape is decreased when (a) regressing a single shape directly from data (such as in IGR [1] Section 6), and (b) regressing a shape using an auto-decoder.**
> **A:** Thank you for this suggestion: the paper indeed benefits from such an experiment. We added an experiment to the revised paper (see section 4.3), addressing the question of regressing a single shape. As can be learned by this experiment, SALD indeed enjoys better sample complexity than SAL, especially for low sample sizes. Our plan is to include also the experiment on regressing a shape using an auto-decoder in the next revision.
>
> **Q: Showing that global minima to SAL may satisfy the minimal surface property is indeed quite interesting. I do feel however that the claim in the paper regarding this is a bit oversold….I feel the contribution should be rephrased to something along the lines of "We give empirical evidence and theoretical motivation that minimizers of SAL-type losses produce solutions satisfying the minimal surface property".**
> **A:** We accept this comment and have edited the abstract and the introduction accordingly.
>
>
> **Q: I imagine that computing losses on gradients of networks is quite expensive. How much is the increase in runtime compared to the gains in accuracy?**
> **A:** Thank you for this comment. Indeed there is an additional computational cost for calculating the gradients of the network. We added a new section in the supplementary providing computational timings and memory footprint (see section A.2.3).
>
> [1]: implicit geometric regularization for learning shapes. InProceedings of Machine Learning and Systems 2020, 2020.

---

> > ### Author Response · Authors · 2020-11-24
> > **Response to reviewer4**
> >
> > Thank you for suggesting to test the sample complexity hypothesis. We have uploaded a revised version of the paper which now includes both experiments you suggested: shape reconstruction, and latent shape reconstruction using a trained auto-decoder. The details are in section 4.3.

---

> ### Comment · AnonReviewer4 · 2020-11-24
> **Revision looks good**
>
> Thank you for the additional experiments and for the updated draft. I feel like they improve the quality of the paper overall. Given these changes, I feel the paper is in a good enough state to accept for publication.

---

### Official Review · AnonReviewer1 · 2020-10-28
**Good problem, weak motivation, and issues in experiment design.**

**Rating:** 6
**Confidence:** 5

**Review:**

This paper studies how to generate meshes from raw point clouds. In particular, this paper proposes a framework which is built on top of recent "sign agnostic learning (SAL)" work. Compared to SAL, this work adds a gradient penalty term, which encourages the derivative consistency. The problem studied in this paper is important, however, the proposed method is very incremental and has several motivation issues. I summarize the pros and con as follows.

Pros:
1. The idea of using gradient penalty to learn "sharp" signed distance function seems convincing. In Figure 4, the proposed method preserves sharp features compared to its counterpart SAL.
2. This paper presents a theoretic intuition why SALD works -- under uniform distribution assumption, SALD finds the global minimum.

Cons:
1. My biggest concern is the motivation to learn sign distance function from its unsigned observations. For data (ShapeNet and FAUST) used in this paper, signed distances are immediately available -- one can easily convert a mesh to its implicit representation. To me, learning signed distance function (as DeepSDF does) is more convincing since the direct supervision is available. So why does this method bother to learn the proxy objective (unsigned distance function)?
2. Following 1, the most obvious application of this paper would be learning signed distance function when the distances are not available -- the input is either LiDAR scan or depth image. In that case, if the paper can reconstruct realistic 3D models, it will be much stronger.
3. To some extent, this paper uses neural networks to learn sign priors from data. There are multiple existing works on this direction which this paper doesn't mention (or briefly mentions but doesn't compare to). E.g, "Deep geometric prior for surface reconstruction" and "Point2Mesh: A Self-Prior for Deformable Meshes". The paper should at least explain the differences of the tasks if it doesn't compare to them.
4. In the implementation detail, the paper says it uses a similar architecture to DeepSDF in the auto-decoding case. However, the method shows improvements over DeepSDF. This seems impossible given that DeepSDF learns from direct signed distance supervision. So I am wondering if this is due to model size difference. I'd like to see more comparisons to DeepSDF under exactly the same model capacity.

---

> ### Author Response · Authors · 2020-11-20
> **Response to reviewer1**
>
> **Q: My biggest concern is the motivation to learn sign distance function from its unsigned observations. For data (ShapeNet and FAUST) used in this paper, signed distances are immediately available -- one can easily convert a mesh to its implicit representation.**
> **A:** We respectfully disagree. In ShapeNet many of the models are non-manifolds with inconsistent normals orientation and computing the signed distance is a non-trivial task. For example, DeepSDF [1] computes the signed distance supervision using a rendering procedure which provides only approximation to the signed distance function and suffers from several drawbacks such as failure in presence of holes and occluded and invisible areas (e.g., the cars’ interior in figure 1). The data used in the DFaust experiment consists of raw scans. These raw scans have many “real-life” defects such as: holes, ghost geometry and noise. Also in this case, computing a signed distance function is rather challenging, see e.g., [2] figure 6  that demonstrates attempts to directly compute the SDF from the raw data.
> To summarize, computing SDF directly from raw data with holes, noise, and occluded parts is a highly non-trivial problem which is at the heart of the surface reconstruction field.
> Another drawback of the DeepSDF approach is based on its  “two stage solution”:  First, Infer the reconstruction individually for each shape; and only then learn a shape space from the extracted 3D supervision. Note that in the first stage each surface is considered **independently** from all other surfaces in the dataset. An advantage in SALD is the ability to learn the signed representations and the shape space **together**.
>
>
> **Q: There are multiple existing works on this direction which this paper doesn't mention (or briefly mentions but doesn't compare to). E.g, "Deep geometric prior for surface reconstruction" and "Point2Mesh: A Self-Prior for Deformable Meshes".**
> **A:** Deep geometric prior for surface reconstruction is a surface reconstruction method, based on an Atlas parametrization of a surface, that was not used for shape space learning. We added a citation to Point2Mesh in our revised previous work. However, Point2Mesh is another surface reconstruction method not adapted for shape space learning. Thus, we find these works are not natural baselines to our experiments.  More relevant to work is AtlasNet which is also a parametrization based method. However, a comparison of SAL versus AtlasNet is already done in [2] , establishing AtlasNet as inferior to SAL (which is inferior to SALD, as we show in this paper) at the task of shape space learning on the D-Faust raw scans dataset. The relevant details are mentioned in the paper (see the first paragraph in section 4.2).
>
> **Q: In the implementation detail, the paper says it uses a similar architecture to DeepSDF in the auto-decoding case. However, the method shows improvements over DeepSDF. This seems impossible given that DeepSDF learns from direct signed distance supervision. So I am wondering if this is due to model size differences. I'd like to see more comparisons to DeepSDF under exactly the same model capacity.**
> **A:** All methods in the experiment section: DeepSDF, IGR, SAL, SALD use the **exact same** architecture of 8 layers MLP, with 512 hidden units. The only difference is that SALD use smoothed-ReLU (Softplus) instead of ReLU activation for continuous gradient computations. The improvement in reconstruction quality of SALD with respect to DeepSDF can be attributed to the following properties: i) SALD learns directly on the input shape, containing occluded parts, whereas DeepSDF uses approximated signed distance supervision derived only from visible parts. ii) DeepSDF does not exploit normal information explicitly. In fact, many of the shapes in ShapeNet have inconsistent normals orientation, a challenge alleviated with the SALD loss.
>
> [1]: DeepSDF:  Learning continuous signed distance functions for shape representation.  InThe IEEEConference on Computer Vision and Pattern Recognition (CVPR), June 2019.
> [2]: SAL:   Sign  agnostic  learning  of  shapes  from  raw  data.    InIEEE/CVF Conference on Computer Vision and Pattern Recognition (CVPR), June 2020.

---

### Official Review · AnonReviewer3 · 2020-11-04
**A good paper addressing an important problem.**

**Rating:** 8
**Confidence:** 4

**Review:**


This paper presents SALD, a new type of implicit shape representation that, in addition to predicting the signed distance function, aligns the gradients of the distance function with that of the neural distance field. The resulting algorithm, for example, has improved approximation power and better preserves the sharp features than the ancestor SAL (sign agnostic learning). The formulation is such that the architecture can consume raw point clouds.

STRENGTHS

This paper certainly speaks to me. First of all, learning implicit representations directly from raw point clouds can allow for interesting applications such as better generative models or efficient 3D reconstruction networks. The approach is very sensible. In fact, aligning gradients of the implicit surface with the ones of the data is not a new idea and has been done for instance in quadric fitting:
* Birdal, T., Busam, B., Navab, N., Ilic, S., & Sturm, P. (2019). Generic primitive detection in point clouds using novel minimal quadric fits. IEEE transactions on pattern analysis and machine intelligence, 42(6), 1333-1347.
* Tasdizen, T., Tarel, J. P., & Cooper, D. B. (1999, June). Algebraic curves that work better. In Proceedings. 1999 IEEE Computer Society Conference on Computer Vision and Pattern Recognition (Cat. No PR00149) (Vol. 2, pp. 35-41). IEEE.

[the paper might benefit from including those especially because it has related work sections called 'primitives' and 'implicit representations'.].

This is not a drawback but just the opposite: there is a strong prior evidence that such approaches are useful. I also like that the authors spend a reasonable amount of effort for theoretical analysis. Though, I believe that this can be extended to more realistic scenarios (as the authors aptly explained in the limitations).


WEAKNESSES / ISSUES

- In addition to aligning the gradients, many works benefit from constraining the gradient norm of the implicit function be |\nabla| = 1. See for instance:
* Slavcheva, Miroslava, et al. "Killingfusion: Non-rigid 3d reconstruction without correspondences." Proceedings of the IEEE Conference on Computer Vision and Pattern Recognition. 2017.

Can we think of a similar approach here? Could the paper show some ablations with regularizers concerning the gradient norm?

- Nowadays, the use of implicit 3D representations is omnipresent. In the evaluations, would it be possible to compare against the variants of DeepSDF (e.g. Curriculum DeepSDF or MetaSDF etc.)? With that, it might also be nice to include some more qualitative results in the supplementary.

- Would it be possible to include additional real objects that are non-humans? This might involve for instance cars in an autonomous driving scenario.

- Some discussions on the following aspects could be valuable for the reader: (i) What would be a good suggestion to handle thin-structures? It seems to be a common issue among many SDF-like methods. (ii) The use of raw point sets is good, but such data usually come partially observed. Could this method support partial observations? If not, could there be workaround?

- The Chamfer distance and the variations thereon are obviously not well suited to assess the accuracy of the deep implicit representations. This creates an urge for better quantitative metrics, maybe the data driven ones. For the future, I would strongly suggest thinking about those to have more meaningful evaluation data.

- Some minor remarks:
* Can we already compare D and D' and give an intuition about what they might refer to at the place they are first defined?
* "they strives to" -> they strive to
* "tested SALD ability" -> tested SALD's ability
* "the surfaces produces" -> "the surfaces produced"

---

> ### Author Response · Authors · 2020-11-20
> **Response to reviewer3**
>
> **Q: In fact, aligning gradients of the implicit surface with the ones of the data is not a new idea and has been done for instance in quadric fitting**
> **A:** Thank you for pointing out these references, we have added them to our revised previous work section.
>
> **Q: In addition to aligning the gradients, many works benefit from constraining the gradient norm of the implicit function be $|\nabla f| = 1$. Can we think of a similar approach here?**
> **A:**  Thank you for this interesting question. We think that the main difference between SALD to gradient norm penalty methods (such as IGR [1]) relates to the way the learned implicit function completes missing parts. The SALD loss explicitly regularizes for solutions to possess minimal surface area, whereas in IGR the regularization leads to constant curvature-like solutions. For example, in the D-FAUST experiment (section 4.2), we see that the SALD approach is more suitable for completing missing parts in the areas of the human feet (see figure 8 in the paper).  We revised the paper to include more details about this issue: see the paragraph about the minimal surface property in section 3 and figure A.1 in the appendix.
> Lastly, in the revised conclusions section we added a comment that incorporating sign-agnostic losses with gradient norm penalty is an interesting future work direction, potentially combining the advantages from both methods.
>
> **Q: Compare against the variants of DeepSDF (MetaSDF and Curriculum DeepSDF).**
> **A:** We believe these methods solve different problems than SALD, and therefore do not serve as natural baselines.   Curriculum DeepSDF is a method suggesting a weighted signed distance regression, where the weights are extracted based on 3D supervision (such as the sign information). As our method addresses the problem of learning signed solutions **without** 3D supervision, it is not immediately clear how to incorporate this into the sign-agnostic framework.
> MetaSDF is a recent approach for shape space learning, based on ideas from Meta Learning. In our paper, we incorporate SALD  in Auto-Decoder (AD) and Variational Auto-Encoder (VAE) which are two other state-of-the-art shape space learning architectures. Indeed, SALD can also be incorporated in MetaSDF. Although the choice of the shape space learning architecture and method is a very interesting research question, we feel it is somehow orthogonal to SALD contribution, concentrating on the reconstruction loss rather than shape space generalization. We therefore leave this to be investigated in future works.
>
> **Q: Would it be possible to include additional real objects that are non-humans?**
> **A:** First, please note that ShapeNet models are non-human models that are very often modeled as triangle soups, that is, not manifolds and possess inconsistent normals. In that aspect the ShapeNet experiments in the paper provide real non-human objects. As to **raw scans** of non-human objects - we are not aware of such freely available large-scale dataset and we agree the community would benefit tremendously from such a dataset.
>
>
> **Q: discussions on the following aspects could be valuable for the reader: (i) What would be a good suggestion to handle thin-structures? (ii) The use of raw point sets is good, but such data usually come partially observed. Could this method support partial observations?**
> **A:**  (i) We added a discussion to the revised paper about  learning thin-structures with implicit neural representation (see section 4.4 in the paper). (ii) SALD is a suitable method to tackle partial observations up to some extent due to its minimal surface property. For instance, many of the D-Faust raw scans contain holes which are gracefully completed by SALD. More challenging scenarios (i.e., large missing parts) should probably be treated with appropriate shape space regularization, which is again a very interesting research direction but outside the scope of the current paper.
>
>
> **Q: Can we already compare D and D' and give an intuition about what they might refer to at the place they are first defined?**
> **A:** We added such an explanation in the revised paper.
>
> [1]: implicit geometric regularization for learning shapes. InProceedings of Machine Learning and Systems 2020, 2020.

---

### Official Review · AnonReviewer5 · 2020-11-05
**SALD review**

**Rating:** 8
**Confidence:** 3

**Review:**

This paper is based on the "sign agnostic learning" (SAL) method for capturing signed distance functions with neural networks. It extends this method by incorporating derivative information, which interestingly can likewise be handled in a sign agnostic manner. (Maybe I missed this somewhere, but if the derivatives are sign agnostic, couldn't it happen that the inside is positive? Did the authors encounter that in some cases?)

The paper presents and motivates this extension together with an additional theoretical insight about the minimal surface property of SAL and SALD. In line with SAL, the paper presents a nice variety of results for shapes from different shape databases. The quantitative results are also convincing. It's interesting to see the substantial difference between the VAE and AD architectures. For the comparison with SAL it's good to see the direct improvements from the derivative loss with a VAE.

The paper leans heavily on SAL, and the change in terms of the overall method seems to be fairly small. Nonetheless, I think it's an interesting insight that the sign agnostic derivatives can be included in this way, and I found it interesting to see how much they improve the results.

Given that learning signed distance functions is a very active topic, and a very useful building block for a variety of adjacent works that use learned SDFs, the proposed SALD approach seems like a very nice advancement of the state of the art.

So, overall, I really liked the paper. Figure 2 alone is impressive, and makes a good case for the method. Together with the nice presentation and set of results I think this paper makes for a very good addition to ICLR.

---

> ### Author Response · Authors · 2020-11-20
> **Response to reviewer5**
>
> **Q: Maybe I missed this somewhere, but if the derivatives are sign agnostic, couldn't it happen that the inside is positive? Did the authors encounter that in some cases?**
> **A:** The SALD loss does not encourage a specific sign inside of the shape. Namely, both solutions (i.e., negative inside and positive outside, and vice-versa) are local minima. However, each of these solutions is stable. That is, continuously moving from one signed solution to another (during optimization) would yield a significant increase in the SALD loss. In practice, the solutions SALD converge to are the ones where the sign of the inside of the shape is always negative. This is a result of the geometric initialization scheme used for SALD (see figure 3 in [1]).
>
> [1]:SAL:   Sign  agnostic  learning  of  shapes  from  raw  data.    InIEEE/CVF Conference on Computer Vision and Pattern Recognition (CVPR), June 2020.

---

### Author Response · Authors · 2020-11-24
**Thank you for the reviews**

With this discussion period coming to an end, we would like to thank the reviewers for their constructive suggestions and remarks; we really feel it improved the paper. We have uploaded another revised version of the paper which includes a sample complexity experiment.

---

### Decision · Program_Chairs · 2021-01-07
**Final Decision**

**Decision:**

Accept (Poster)

**Comment:**

Congratulations!  The reviewers unanimously viewed this work positively and were in favor of acceptance to ICLR.

While the current revision already addresses many reviewer concerns, it may be worth adding some of the datasets pointed out by R3 or comparing to some of the papers suggested by R1.